# Focal Uptake in the Sternum on ^18^F-FDG-PET/CT Caused by G-CSF Therapy after Chemotherapy Mimicking Bone Metastasis of Breast Cancer

**DOI:** 10.3390/diagnostics12102308

**Published:** 2022-09-25

**Authors:** Kumiko Hayashi, Tomoyuki Fujioka, Masatake Hara, Yuichi Kumaki, Goshi Oda, Emi Yamaga, Mio Mori, Iichiroh Ohnishi, Kazunori Kubota, Tsuyoshi Nakagawa

**Affiliations:** 1Department of Breast Surgery, Tokyo Medical and Dental University Hospital, Tokyo 113-8510, Japan; 2Department of Diagnostic Radiology, Tokyo Medical and Dental University Hospital, Tokyo 113-8510, Japan; 3Division of Surgical Pathology, Tokyo Medical and Dental University Hospital, Tokyo 113-8510, Japan; 4Department of Radiology, Dokkyo Medical University Saitama Medical Center, Saitama 343-8555, Japan

**Keywords:** ^18^F-fluorodeoxyglucose positron-emission tomography, granulocyte colony stimulating factor, breast cancer, bone marrow, chemotherapy

## Abstract

A woman in her 70s was diagnosed with left breast cancer and left axillary lymph node metastasis by an ultrasound-guided biopsy. ^18^F-FDG-PET/CT showed strong FDG accumulation in the tumor in the left breast and a left axillary lymph node. Neoadjuvant chemotherapy (NAC) was administered in combination with a G-CSF injection to prevent febrile neutropenia. The post-treatment ^18^F-FDG-PET/CT showed the disappearance of the left breast tumor and left axillary lymph node and revealed a solitary new area of strong FDG accumulation in the sternum. To rule out the possibility of sternal metastasis, a sternal biopsy was performed at the same time as surgery, which revealed no malignant findings. Although very rare, focal uptake on ^18^F-FDG-PET/CT performed after anticancer drug therapy with G-CSF may mimic a solitary bone metastasis. A bone biopsy may be a useful technique to avoid an immediate misdiagnosis of bone metastasis.

## 1. Introduction

In breast cancer treatment, **^18^**F-fluorodeoxyglucose positron-emission tomography/computed tomography (**^18^**F-FDG-PET/CT) is used mainly for staging and surveying distant metastases, and also to determine the efficacy of drug therapy [1,2]. Among drug therapies, anthracyclines are frequently used as key drugs in anticancer therapy, especially in breast cancer. As pancytopenia is an almost inevitable side effect of these drugs, a granulocyte colony stimulating factor (G-CSF) injection is usually used in combination with anticancer therapy as a means of preventing febrile neutropenia.

Symmetric accumulation in the red pulp is commonly observed on **^18^**F-FDG-PET/CT after a G-CSF injection [3]. In the present case, however, a single localized region of strong FDG accumulation was seen in the sternum. To the best of our knowledge, this is the first report of this extremely rare appearance.

Here we report the case of a single localized region of strong FDG uptake that appeared in the sternum on PET/CT following G-CSF therapy in NAC for breast cancer, which was confirmed by a bone biopsy as a focal FDG uptake mimicking the bone metastasis of breast cancer.

## 2. Case Presentation

A woman in her 70 s (gravida 3, para 3) presented at our hospital for uterine tumor surgery. She had no relevant medical or family history. Preoperative screening CT revealed a left breast tumor and an enlarged left axillary lymph node, so she was referred to the breast surgery department. Mammography showed an irregular mass with an indistinct margin in the upper left breast (Figure 1a). MRI revealed an irregular mass in the left breast that showed contrast enhancement with a fast-washout pattern (Figure 1b). Ultrasonography showed an irregular hypoechoic tumor in the left breast (Figure 1c) and an enlarged lymph node with a thickened cortex in the left axilla (Figure 1d). 

Ultrasound-guided biopsy of the mass yielded a diagnosis of left breast cancer (invasive ductal carcinoma, estrogen receptor [ER], Allred score 8; progesterone receptor [PR], Allred score 2; human epidermal growth factor receptor [HER2], score 3+; Ki-67, 10.5%, nuclear grade, (1) (Figure 2). The left axillary lymph node was class V cytologically. PET/CT showed **^18^**F-FDG uptake in the left breast mass (Figure 3A(a,b): yellow arrow), left axillary lymph node (Figure 3A(a,c): red arrow), and in a uterine tumor (Figure 3A(a): white arrow). No other distant metastasis was observed (Figure 3A(d)).

The patient underwent total hysterectomy and bilateral oophorectomy. The uterine tumor was pathologically confirmed as low-grade myometrial sarcoma, and the breast cancer was staged as T2N1M0 Stage IIB. She then received four courses of an AC (doxorubicin + cyclophosphamide) regimen and 4 courses of a PER + HER + DOCE (trastuzumab + pertuzumab + docetaxel) regimen as NAC for breast cancer. On the day after each drug treatment, a subcutaneous injection of G-CSF (pegfilgrastim 3.6 mg) was administered to prevent febrile neutropenia. A PET/CT scan was performed 15 days after the last injection of G-CSF in this case. The PET/CT showed no accumulation in the left breast or left axillary lymph node after NAC (Figure 3B(a–c)); however, a new solitary focus of strong FDG accumulation was seen, localized to the sternum (Figure 3B(a,d): blue arrow). As the possibility of sternal metastasis, sternal tumor, and hematopoietic tumor could not be ruled out, a sternal biopsy was performed at the time of the total left mastectomy and axillary lymph node dissection.

Postoperatively, there was no tumor in the left breast or axillary lymph node pathologically, and the chemotherapy response was pCR (treatment response grade 3). The sternal biopsy revealed normal bone marrow and no malignancy (Figure 4). The localized FDG accumulation in the sternum on **^18^**F-FDG-PET/CT was caused by the G-CFS preparation. The patient is currently receiving a trastuzumab + pertuzumab (PER + HER) regimen as adjuvant therapy and has had no recurrence.

## 3. Discussion

We experienced a case of a highly localized single FDG accumulation on PET/CT that was found in the sternum after chemotherapy with G-CSF. Despite the good response of the primary tumor to chemotherapy, the appearance of the new accumulation in the sternum made it difficult to decide the treatment strategy.

After the injection of G-CSF, FDG usually accumulates symmetrically in the bone marrow due to increased hematopoietic activity of the granulocytic system, which in turn increases the bone marrow glucose metabolism [4]. There are several previous reports of localized FDG accumulations resembling multiple bone metastases [5,6,7], but this is the first report of a case of single, localized FDG accumulation in the sternum. The European Association of Nuclear Medicine (EANM) procedure guidelines recommend that a PET-CT scan be performed at least 2 weeks after the G-CSF injection. In our case, however, an abnormal accumulation in the sternum was observed in spite of the 15 days interval [8].

As the subsequent treatment plan may differ greatly depending on the presence or absence of bone metastasis, bone metastasis should be staged (or diagnosed) after confirming the history of the G-CSF injection and bone biopsy should be considered if necessary, rather than simply assuming metastasis when localized FDG accumulation is detected on **^18^**F-FDG-PET/CT. Unless it is urgent, we may consider waiting for PET/CT for a few more weeks after the G-CSF injection in order to evaluate metastases accurately.

## 4. Conclusions

A G-CSF injection may cause a localized accumulation resembling bone metastases on **^18^**F-FDG-PET/CT.

## Figures and Tables

**Figure 1 diagnostics-12-02308-f001:**
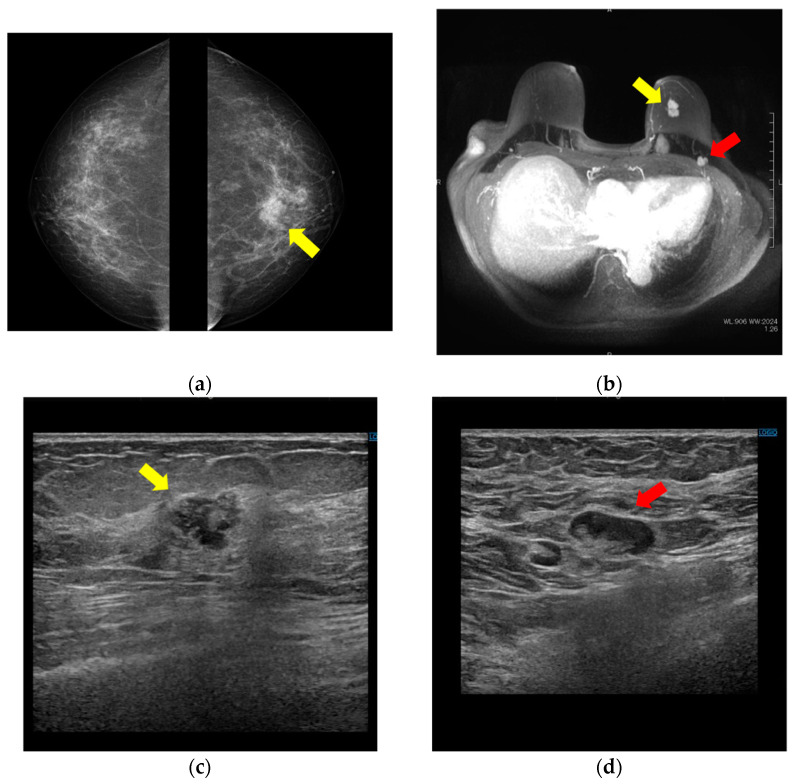
Breast mass and axillary lymph node on mammography, MRI, and ultrasound. (**a**) Bilateral mediolateral oblique mammogram demonstrates a 20 mm irregular mass with indistinct margin in the upper area (yellow arrow). (**b**) MRI shows a 29 mm irregular mass that showed contrast enhancement with fast-washout pattern in the upper area of the left breast (yellow arrow) and enlarged lymph nodes in the left axilla (red arrow). (**c**) Ultrasonography reveals a 17 mm irregular hypoechoic tumor in the upper area of the left breast (yellow arrow) and (**d**) enlarged lymph nodes with thickened cortex in the left axilla (red arrow).

**Figure 2 diagnostics-12-02308-f002:**
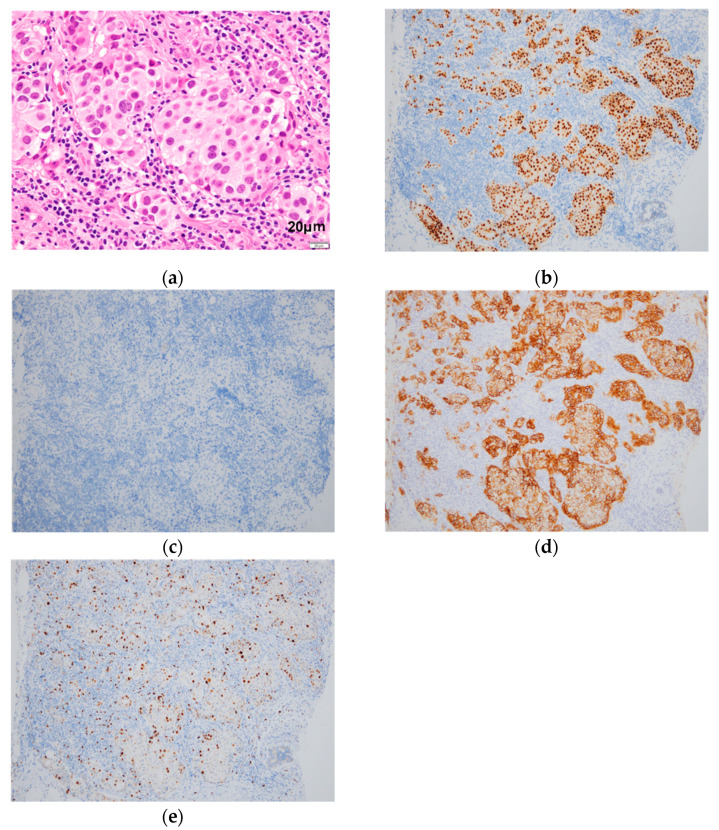
Pathology of breast lesion tissue sampled by ultrasound-guided biopsy. Histological examination reveals invasive ductal carcinoma, scirrhous type. HE staining (**a**); Immunostaining shows estrogen receptor [ER], Allred score 8 (**b**); progesterone receptor [PR], Allred score 2 (**c**); human epidermal growth factor receptor [HER2], score 3+ (**d**); and Ki-67, 10.5% (**e**).

**Figure 3 diagnostics-12-02308-f003:**
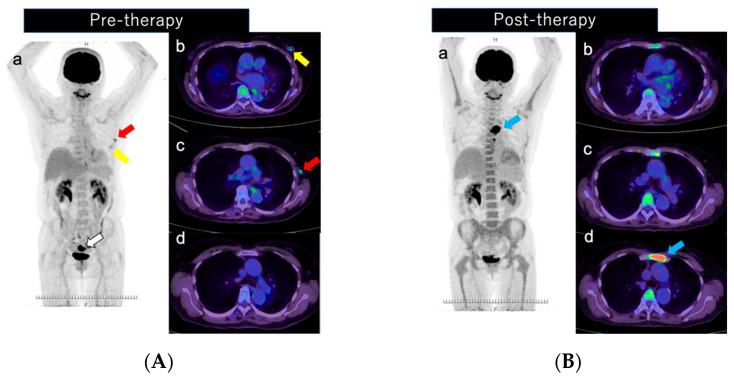
(**A**) Pre- and (**B**) post-therapy PET/CT imaging. (**A**) Pre-therapy imaging shows **^18^**F-FDG uptake in the left breast mass (SUV max, 3.0; (a,b): yellow arrow), a left axillary lymph node (SUV max, 4.2; (a,c): red arrow), and a uterine tumor (SUV max, 25.3; (a): white arrow). No other distant metastasis was observed (d). (**B**) Post-therapy imaging shows no abnormal uptake in the left breast or left axillary lymph node after NAC (a–c). There is strong localized FDG accumulation of a solitary new lesion in the sternum (SUV max, 30.6; (a,d): blue arrow).

**Figure 4 diagnostics-12-02308-f004:**
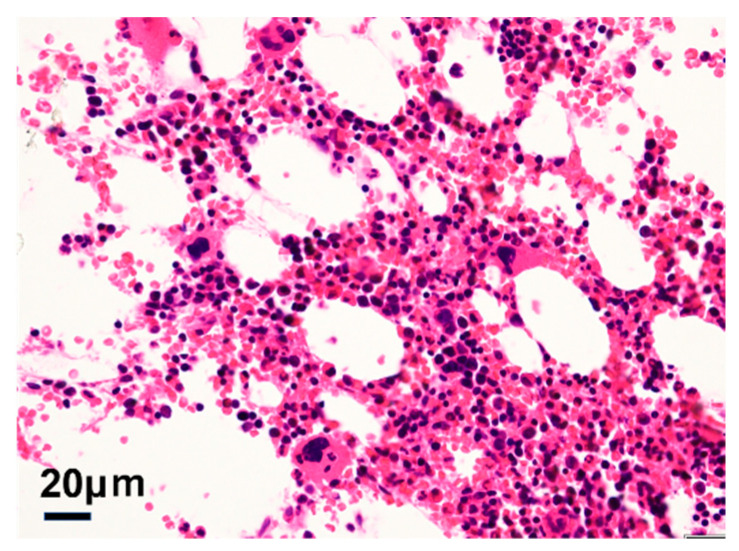
Pathology of bone marrow sampled at the same time as the breast cancer surgery. Histological examination of the bone biopsy tissue shows no malignant findings.

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
