# Peer review of "Focal Uptake in the Sternum on 18F-FDG-PET/CT Caused by G-CSF Therapy after Chemotherapy Mimicking Bone Metastasis of Breast Cancer"

_diagnostics, 2022, doi:10.3390/diagnostics12102308_

Round 1

Reviewer 1 Report

The paper illustrates the case of a female patient affected by breast cancer and ipsilateral axillary lymph node metastases that was submitted to FDG PET CT before and after NAC and G-CSF injection.

In the post-treatment PET study, a focal uptake in sternum was evidenced mimicking bone metastasis that was ruled out by biopsy.

In cancer patients treated with chemotherapy and G-CSF a highly variable increase in bone marrow activity was noted, and increased activity persisted for up to 3 weeks after discontinuation of G-CSF treatment. Thus, in order to avoid FDG PET misinterpretation at the level of the bone (FDG uptakes mimicking bone metastases), it is generally suggested to perform FDG PET approximately 1 month after CSF discontinuation. This recommendation should be added in the Introduction section.

On the basis of the above considerations, it is very important to known how many time has passed since G-CSF treatment and post-therapy FDG PET in this specific case. This information is not reported in the manuscript while it would be of great importance for the discussion and conclusions.

Thus, I suggest that the authors add this information and review the discussion and conclusions of the manuscript accordingly.

Reviewer 2 Report

1. Have reporters about G-CSF injection caused false positive findings on 18F-FDG-PET/CT?

2. Sternum lesion was possible ......, add in the discussion.

3. All 18F-FDG-PET/CT changes into 18F-FDG-PET/CT

4. References, recheck the type.

Round 2

Reviewer 1 Report

No further comments. The authors answered the questions asked.